# Introducing heat-not-burn tobacco improves hematocrit and cigarette smoking-related symptoms in patients with smokers' polycythemia and polycythemia vera

Kazuhide Iizuka[1,2,3*], Yoshikazu Iizuka[2], Shun Ito[2], Toshihide Endo[2], Hironao Nukariya[2], Yuichi Takeuchi[2], Takashi Koike[2], Kazuya Kurihara[2], Hiromichi Takahashi[2], Masaru Nakagawa[2], Takashi Hamada[2], Shimon Ohtake[2], Noriyoshi Iriyama[2], Katsuhiro Miura[2], Tomohiro Nakayama[1], Yoshihiro Hatta[2], Hideki Nakamura[2], Norio Komatsu[4,5]

**1** Division of Laboratory Medicine, Department of Pathology and Microbiology, Nihon University School of Medicine, Itabashi-ku, Tokyo, Japan, **2** Division of Hematology and Rheumatology, Department of Medicine, Nihon University School of Medicine, Itabashi-ku, Tokyo, Japan, **3** Division of Hematology, Tokoro Memorial Hospital, Atami city, Shizuoka, Japan, **4** Department of Hematology, Juntendo University Graduate School of Medicine, Bunkyo-ku, Tokyo, Japan, **5** Pharmaessentia Japan KK, Minato-ku, Tokyo, Japan

* iizuka.kazuhide@nihon-u.ac.jp

## Abstract

Cigarette smoking induces relative and absolute erythrocytosis (smokers' polycythemia). In patients with smokers' polycythemia complicated by chronic obstructive pulmonary disease, the incidence and mortality rate of pulmonary embolism increase. Therefore, improving erythrocytosis by smoking cessation is important; however, smoking cessation is often difficult to achieve. This study investigated the influence of introducing heat-not-burn tobacco in patients with smokers' polycythemia. Fifteen smokers with erythrocytosis were diagnosed with smokers' polycythemia (n = 13) or polycythemia vera (n = 2) groups. The patients selected a switch to heat-not-burn (HNB) tobacco or smoking cessation, and the subsequent changes in hematological data and symptoms were evaluated. Eight patients with smokers' polycythemia and two with polycythemia vera selected a switch to HNB tobacco, and the other five patients with smokers' polycythemia selected smoking cessation. In both the HNB tobacco and smoking cessation groups, all patients showed improved hematocrit (Hct) and sputum volume and ameliorated numbness, headache, and vertigo. In the patients with smokers' polycythemia, Hct of the HNB tobacco group was equivalent to that in the patients with smoking cessation group (47.51 ± 3.48% vs. 45.63 ± 2.79%, P = 0.605). Introduction of HNB tobacco may be useful for reducing erythrocytosis in smokers for whom smoking cessation is difficult.

**Data availability statement:** All relevant data are within the paper and its Supporting Information files.

**Funding:** This study was financially supported by JSPS KAKENHI in the form of a grant (21K16253) received by KI. This study was also financially supported by Kaltech Corporation in the form of contracted research funding received by KI. The funders had no role in study design, data collection and analysis, decision to publish, or preparation of the manuscript.

**Competing interests:** The authors have declared that no competing interests exist.

## Introduction

Carbon monoxide (CO) plays an important role in the onset of smokers' polycythemia [1,2]. Smoking not only causes hypoxemia by developing COPD, but also CO contained in cigarette smoke has a strong affinity for hemoglobin (Hb) and induces tissue hypoxia [2–4]. In addition, the mRNA expression of erythropoietin receptor-1 (EpoR-1) is upregulated in smokers [5]. In fact, Patients with smoking polycythemia tend to have low erythropoietin levels [5,6]. These mechanism causes absolute polycythemia. Furthermore, CO enhances vascular permeability, causing relative polycythemia through a decrease in the circulating blood volume [1,4]. In fact, HbCO is positively correlated with the number of cigarettes smoked per day [7] and is also positively correlated with Hb [8]. In contrast to a 1993 report in which polycythemia vera (PV) was not confirmed by testing for *JAK2* (V617F and exon 12) mutations, [9] recent studies have shown that secondary polycythemia and PV have similar rates of thromboembolism, [10] suggesting that secondary polycythemia also warrants intervention to reduce the risk of thrombosis in this cohort. In patients with smokers' polycythemia complicated by chronic obstructive pulmonary disease (COPD), the incidence of pulmonary embolism and low-risk (pulmonary embolism severity index; PESI score ≤ 85) pulmonary embolism-related mortality rate increase [11]. Therefore, the need to ameliorate smokers' polycythemia has been emphasized, and increased the need for consultation with a hematologist. However, patients with smokers' polycythemia often struggle to stop smoking, and no effective treatment, other than smoking cessation, has been established. This is the first study to examine whether switching from cigarettes to heat-not-burn (HNB) tobacco improves polycythemia and its concomitant symptoms in smokers. The characteristic of HNB tobacco is no combustion, lower temperature than cigarettes and do not produce smoke containing CO. On the other hand, metal aerosols are generated. Therefore, concerning HNB tobacco, surveys regarding its health effects are insufficient, and its safety is controversial. However, the amelioration of smokers' polycythemia may lead to an improvement in the prognosis of patients with thrombosis or COPD. Herein, we report that switching to HNB tobacco improves polycythemia and subjective symptoms in smokers.

## Methods

### Patients

This study was conducted in accordance with the principles of the Declaration of Helsinki, and the study protocol was approved by the Institutional Review Boards of Nihon University Itabashi Hospital (IRB no: 248−2) and Atami Tokoro Memorial Hospital (IRB no: 20200629). Written informed consent regarding the use of samples and clinical records was obtained from all patients before sample collection. From August 1, 2009, to December 31, 2021, we recruited cigarette smokers who met the WHO 2016 hematocrit (Hct) and Hb diagnostic criteria [12] for PV (Hct > 49% and/or Hb 16.5 g/dL in males; Hct > 48% and/or Hb 16.0 g/dL in females) and who had a history of failing a challenge of smoking cessation in the past. In addition, the dyspnea of all patients was grade I on the Fletcher-Hugh-Jones classification and grade 0 on the MRC (Medical Research Council) scale. No patients were diagnosed with COPD on

chest X-ray. They lived between 20 and 70 m above sea level. Of the 40 patients with polycythemia who were smokers, the cause of polycythemia was investigated and differentiated in 26 patients. Seven patients were excluded owing to obesity (BMI > 35; n = 2), diabetes mellitus (n = 4), or mental stress at work (n = 1).

## Diagnosis

The patient differentiation process is shown in Fig 1. In smokers with polycythemia, causes of secondary polycythemia other than smoking were excluded: obesity (BMI ≥ 35), diabetes mellitus, psychological stress, sleep apnea syndrome, hyperthyroidism (hypertensive patients only). In addition, it was confirmed that erythropoietin was normal or low, and *BCR::ABL* gene fusion and *JAK2* mutations (V617F and exon 12 mutation) were negative. Furthermore, patients who were negative for *JAK2* mutations, whose bone marrow biopsy did not show hypercellularity with trilineage growth, were diagnosed as having smokers' polycythemia.

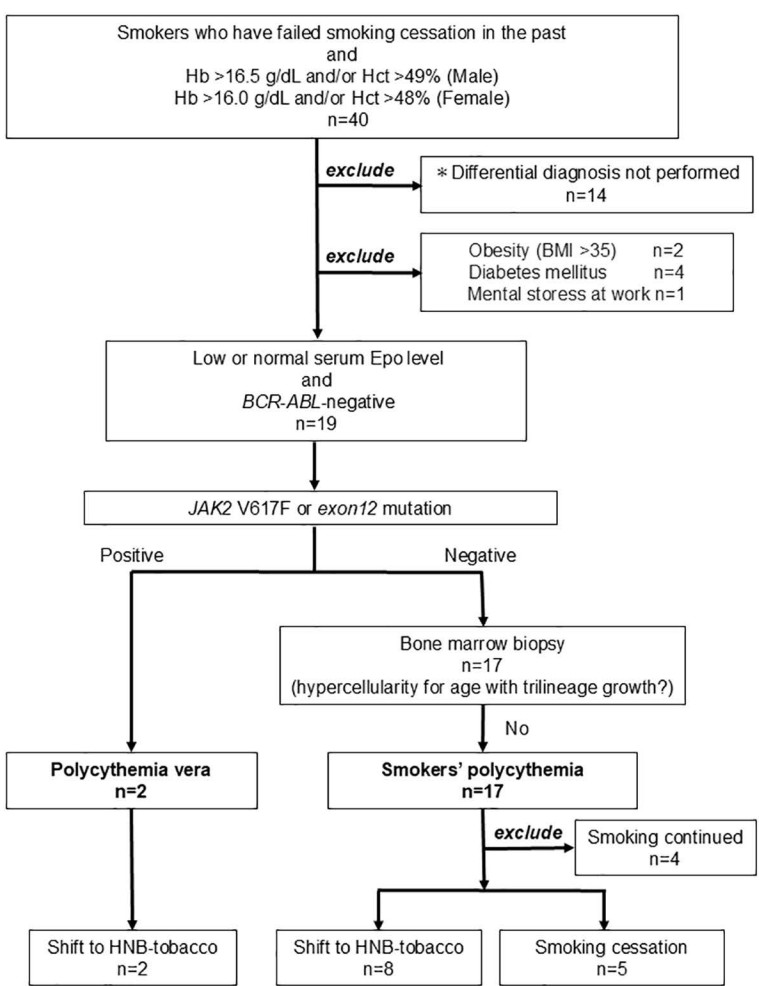

**Fig 1. Differentiation of smokers' polycythemia from polycythemia vera.** After After investigating the etiologies of secondary polycythemia other than smokers' polycythemia, the patients were divided into smokers' polycythemia and polycythemia vera groups according to the WHO 2016 diagnostic criteria for polycythemia vera. *These are patients for whom consent was obtained but some of the tests necessary for differential diagnosis were not performed due to circumstances.

### Gene analysis

*BCR::ABL* gene fusion was assessed using Fluorescence in-situ hybridization at SRL, Inc. (Tokyo, Japan). *JAK2* mutations (V617F and exon12 mutation) were detected using allele-specific polymerase chain reaction (AS PCR) at SRL, Inc. AS PCR uses the same reverse primer and two different forward primers for the normal allele and the mutant allele to compete in the PCR reaction to semi-quantitatively measure the *JAK2*V617F mutation and *JAK2* exon12 mutation, which is a single nucleotide mutation.

### Treatment administration and assessment of outcomes

At the time of participation in this study, patients were recommended to reduce or quit smoking. In patients who could not quit smoking despite this recommendation, HNB tobacco was introduced. Smoking habits were self-recorded and tracked weekly. Two patients had undergone phlebotomy but had discontinued phlebotomy more than three months before entering this study, and there was no resumption of phlebotomy during the observation period. Blood samples were collected before the switch from habitual smoking and at about 2-month intervals after the switch. The patients' white blood cell (WBC) counts, Hb, Hct, D-dimer levels, and changes in symptoms were recorded.

### Statistical analysis

We used R soft for the statistical analysis and graphical representation. Using the Mann–Whitney U test, the baseline features of the patients with smokers' polycythemia were compared between those that shifted to HNB tobacco and those who quit smoking. Habitual smoking alteration-related changes in hematological data were compared using t-tests.

## Results

### Patients

Two patients were *JAK2*V617F positive and were diagnosed with PV. Seventeen patients were diagnosed with smokers' polycythemia, but four did not participate in the study because they continued smoking (Fig 1). Therefore, 13 patients with smokers' polycythemia and two patients with *JAK2*V617F-positive PV were enrolled in this study (Fig 1). Among those with smokers' polycythemia, five selected smoking cessation, and eight selected a shift to HNB tobacco (Fig 1). No significant differences in age, number of cigarettes smoked, or blood count were observed between the 2 groups before the change in smoking habits (Table 1). Of the five patients who chose smoking cessation, four were successful. The patient who could not quit smoking decreased his amount of smoking by half, and this decrease was maintained. In the HNB tobacco group, a complete switch to HNB tobacco was achieved in all eight patients. In the two patients with PV, a switch to HNB tobacco was achieved. In the HNB tobacco group, the number of smoking items per day during the follow-up period was consistently maintained from the start of the study until its completion. For each patient, no changes in medications or treatments were observed. Among the patients with PV, there were no changes in drug administration during the follow-up period.

### Changes in hematological data

The changes in Hct in the patients with smokers' polycythemia are shown in Fig 2. The Hct, WBC count, and symptoms are presented in Table 2. A significant improvement in the Hct was observed in both groups (HNB, p < 0.001; smoking cessation, p = 0.017), while no changes were observed in the WBC count (HNB, p = 0.731; smoking cessation, p = 0.369). The Hct after a switch to HNB tobacco was 47.51 ± 3.48%, and that after the achievement of smoking cessation was 45.63 ± 2.79%; there was no significant difference between the two groups (p = 0.599). Among the patients with smokers' polycythemia, no increase in the D-dimer level was observed (S1 Table) before or after switching to HNB tobacco/smoking cessation. Similarly, a marked improvement in the Hct was observed in the smokers with PV, but not in the WBC count

**Table 1. Comparison of characteristics in patient with smokers' polycythemia before switching their smoking habits.**

| Group Number | | Heat not burn-tobacco | Smoking cessation | |
|---|---|---|---|---|
| | Total 13 | 8 | 5 | P-value |
| Age (years) | | 51.6±12.0 | 60.8±13.7 | 0.171 |
| Smoking amount (items/day) | | 24.4±10.59 | 25.5±8.73 | 0.873 |
| WBC (×10⁹/L) | | 8.09±1.80 | 7.45±1.60 | 0.305 |
| RBC (×10¹²/L) | | 5.53±0.47 | 5.84±0.59 | 0.768 |
| Hemoglobin (g/dL) | | 17.92±1.00 | 18.88±1.62 | 0.354 |
| Hematocrit (%) | | 54.35±2.20 | 56.63±5.73 | 0.755 |
| Platelet count (×10⁹/L) | | 323.8±231.5 | 248.26±105.4 | 0.724 |
| Erythropoietin (mIU/mL) | | 6.64±4.03 | 12.08±14.13 | 0.435 |
| BMI (kg/m²) | | 24.58±3.27 | 23.79±5.66 | 0.622 |
| TSH* (μIU/mL) | | 2.45±1.12 | 2.36±0.91 | 1.000 |
| FT3* (pg/mL) | | 3.17±0.50 | 3.24±0.75 | 1.000 |
| FT4* (ng/dL) | | 1.26±0.11 | 1.24±0.58 | 1.000 |

Values are means±standard deviation of the mean for heat-not-burn (HNB) tobacco and smoking cessation group.

WBC; white blood cell count, RBC; red blood cell count, BMI; body mass index.

\* Only patients with palpitations/ hypertension/ hot flashes were measured (HNB tobacco group n=4, smoking cessation group n=2).

(S2 Table). Of the patients who switched to HNB tobacco, smoking cessation was finally achieved in two of the eight patients with smokers' polycythemia and in one of the two patients with PV. The Hct did not change after the switch from HNB tobacco to smoking cessation in any of the three patients (Fig 2 and S1 Fig).

## Relief of symptoms

All patients with smokers' polycythemia and PV had peripheral arterial disease (PAD)/Buerger's disease-like symptoms (numbness of the distal portion of the extremities, pain, and color changes such as flush), headache, vertigo, cough, pharyngeal discomfort, or an increase in sputum volume. These symptoms improved in all patients after switching to HNB tobacco or smoking cessation (Table 2).

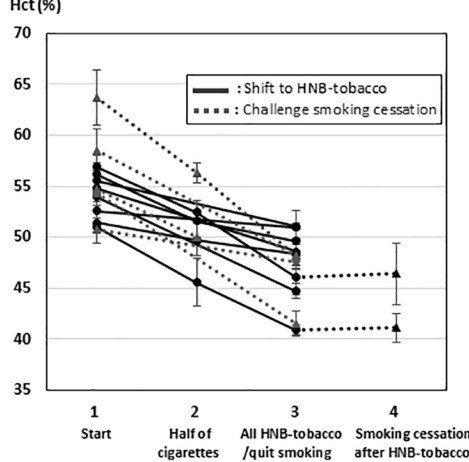

**Fig 2. Changes in the Hct value in patients with smokers' polycythemia.** Solid line: patients that used heat-not-burn (HNB) tobacco; dotted line: patients that achieved smoking cessation. *In 2 patients, complete smoking cessation was achieved after a switch to HNB tobacco.

Table 2. Features of patients with smokers' polycythemia after ≥2 month change to heat not burn (HNB) – tobacco or smoking cessation.

| No. | Patients group | Cigarettes (number/day) | Hct (%) | | | WBC (×10³/μL) | | | Symptom | |
|---|---|---|---|---|---|---|---|---|---|---|
| | | | Start | Half and half* or Half of cigarettes | After shift to HNB tobacco or smoking cessation | start | Half and half* or Half of cigarettes | After shift to HNB tobacco or smoking cessation | Start | After shift HNB tobacco or smoking cessation |
| 1 | HNB tobacco | 40-50 | 56.17±1.91 | 51.57±1.70 | 49.63±0.31 | 9.43±0.21 | 7.50±0.35 | 7.77±0.15 | Headache · Finger pain Reddish hand | none |
| 2 | | 25-30 | 56.90±0.35 | 52.43±1.10 | 46.03±1.65 | 5.80±0.35 | 5.43±0.15 | 5.23±0.25 | Heavy headedness dizziness | none |
| 3 | | 10 | 54.80±0.85 | N/A | 48.57±1.07 | 5.85±0.21 | N/A | 5.63±0.38 | Headache · Tinnitus dizziness | none |
| 4 | | 20 | 51.00±1.59 | 45.55±2.33 | 40.87±0.55 | 6.63±0.70 | 10.85±2.76 | 9.00±0.46 | Headache · Finger pain Reddish hand | none |
| 5 | | 20 | 55.50±1.06 | N/A | 51.05±0.35 | 9.23±0.19 | N/A | 9.45±1.34 | Reddish hand Finger pain | none |
| 6 | | 30-35 | 51.40±0.44 | 49.7±0.28 | 48.30±1.51 | 10.02±0.99 | 10.90±0.57 | 8.40±1.41 | Tinnitus | none |
| 7 | | 20 | 52.55±2.05 | N/A | 50.95±1.63 | 9.90±1.70 | N/A | 9.80±1.77 | Tinnitus | none |
| 8 | | 20 | 53.90±0.46 | N/A | 44.70±0.71 | 7.83±0.45 | N/A | 6.95±0.35 | Headache · Tinnitus | none |
| 9 | smoking cessation | 25-30 | 63.70±2.69 | 56.3±0.99 | 48.20±1.27 | 8.70±1.70 | 9.85±1.70 | 8.85±1.34 | Headache · dizziness · Reddish hand | none |
| 10 | | 20 | 54.33±0.21 | N/A | 41.57±1.15 | 5.73±0.35 | N/A | 10.10±3.40 | Headache · Tinnitus · dizziness | none |
| 11 | | 20 | 58.55±2.05 | N/A | 48.15±0.21 | 5.80±0.71 | N/A | 5.67±1.31 | Reddish hand Finger pain | none |
| 12 | | 20 | 54.73±1.62 | 50.00±1.85 | N/A | 9.15±1.77 | 9.27±1.94 | N/A | Heavy headedness dizziness | none |
| 13 | | 40 | 50.80±0.14 | N/A | 47.60** | 7.85±1.48 | N/A | 8.10** | dizziness · Fatigue | none |

HCT; hematocrit, WBC; white blood cell count N/A: Not applicable data.

*Half and half: Half and half number of HNB tobacco and cigarettes (HNB-tobacco group), Half of cigarettes: Half number of cigarettes (smoking cessation group) **Only one point data.

## Long-term changes in the smoking habit

The median follow-up period after switching the smoking habit was 15 months (range 6–42 months). In the HNB tobacco group, there was no resumption of cigarettes in any patient. On the other hand, two (Patient No. 9 and 13) of the four patients who achieved smoking cessation resumed smoking at a half of the original smoking amount or the same amount (10–12 cigarettes/day, 40 cigarettes/day, respectively) after three months. Patients who returned cigarette consumption to before smoking cessation, their Hct level returned to the same level at the enrollment in this study (No. 9; 59.67±2.00%, No.13; 51.2±0.33%).

## Discussion

In this study, switching from cigarettes to HNB tobacco improved polycythemia and cigarette smoking-related symptoms. Cigarette smoke contains a large amount of CO and the concentration of HbCO is higher in the blood of smokers compared to nonsmokers, causing polycythemia, headache and vertigo related to CO poisoning [1,13–15] The CO content of HNB tobacco smoke is markedly lower than that of cigarette smoke [16,17], which may be the most important factor for improving polycythemia and smoking-related symptoms. Interestingly, switching to HNB tobacco not only improved symptoms of CO poisoning (such as headache and dizziness), but also PAD and Buerger's disease-like symptoms in all patients. The role of nicotine in PAD/Buerger's disease pathogenesis is known, but the involvement of CO has not yet been reported [18,19]. Nicotine levels measured in the aerosol of HNB tobacco are lower than those from cigarettes [20–22]. However, in a rat study, the serum nicotine levels were about 4.5-fold higher in HNB tobacco than in cigarettes [22]; thus, those symptoms may have been relieved through a mechanism not involving nicotine. One possible mechanism is that absolute erythrocytosis activates thrombin [23] and increases platelet P selectin expression [24], which in turn increases the risk of thrombosis. In fact, secondary erythrocytosis has been reported to increase the risk of venous thromboembolism [25] and pulmonary embolism [11]. In this study, none of the patients had high D-dimer levels before switching to HNB tobacco. Therefore, we could not verify whether a switch to HNB tobacco decreases the risk of thrombosis based on the hematological data. However, the fact that absolute erythrocytosis increases the risk of thrombosis suggests the possibility that the amelioration of smokers' polycythemia may have reduced the symptoms of peripheral artery occlusion.

The incidence of arterial thrombosis in patients with PV who smoke is 1.9 times higher than in patients who are non-smokers [26,27], and high Hct levels also increase the risk of thrombosis [28]. In this study, it was intriguing that high Hct levels improved after switching to HNB tobacco without additional treatment in two PV patients. However, due to the small sample size, further expanded studies are needed.

Regarding the risk of secondary carcinogenesis related to cigarettes, cigarette smoking increases the incidence of 17 types of cancer: lung, upper aerodigestive tract (oral cavity, nasal cavity, nasal sinus, pharynx, larynx), esophagus, stomach, pancreas, liver, kidney, lower urinary tract (renal pelvis and urinary bladder), and uterine cervix cancers, myelogenous leukemia, and myeloproliferative neoplasms (including PV) [29–31]. On the other hand, based on DNA methylation of the 17 genes identified as etiological factors for cigarette-related carcinogenesis, seven genes were reported to be significantly less prevalent in HNB tobacco consumers [32]. Nevertheless, HNB tobacco also contains some harmful substances, and the details of its health effects remain to be clarified.

This study suggests that smoker's polycythemia is improved by switching to HNB tobacco. In addition, 1) ameliorated cough/sputum, 2) increased health awareness, and 3) a reduction in smoking volition related to differences in taste between cigarettes and HNB tobacco during smoking were noted and led to smoking cessation in some patients. Although complete smoking cessation is the best treatment, HNB tobacco may be considered as an alternative when cigarette smoking cessation is unsuccessful.

## Supporting information

**S1 Fig. Changes in hematocrit (Hct) in smokers with polycythemia vera.** Solid line: changes in Hct related to a switch to heat-not-burn (HNB) tobacco; dotted line: changes in Hct after a switch to smoking cessation. *In patients with polycythemia vera, neither drug administration nor exsanguination volume changed during follow-up.
(TIF)

**S1 Table. Hematological data before a switch of habitual smoking.**
(DOCX)

**S2 Table. Features of polycythemia vera patients after ≥2 month change to Heat-not-burn (HNB) tobacco.**
(DOCX)

**S1 File. Patients data 1: at the start time.**
(PDF)

**S2 File. Patients data 2: Before and 2 months or more after changing smoking habits".**
(PDF)

## Acknowledgments

We would like to thank Editage (www.editage.com) for English language editing.

## Author contributions

**Conceptualization:** Kazuhide Iizuka.

**Data curation:** Kazuhide Iizuka.

**Formal analysis:** Kazuhide Iizuka.

**Funding acquisition:** Kazuhide Iizuka.

**Investigation:** Kazuhide Iizuka.

**Methodology:** Kazuhide Iizuka, Yoshikazu Iizuka.

**Project administration:** Kazuhide Iizuka.

**Supervision:** Yoshikazu Iizuka, Yoshihiro Hatta.

**Validation:** Kazuhide Iizuka.

**Visualization:** Kazuhide Iizuka.

**Writing – original draft:** Kazuhide Iizuka.

**Writing – review & editing:** Kazuhide Iizuka, Yoshikazu Iizuka, Shun Ito, Toshihide Endo, Hironao Nukariya, Yuichi Takeuchi, Takashi Koike, Kazuya Kurihara, Hiromichi Takahashi, Masaru Nakagawa, Takashi Hamada, Shimon Ohtake, Noriyoshi Iriyama, Katsuhiro Miura, Tomohiro Nakayama, Yoshihiro Hatta, Hideki Nakamura, Norio Komatsu.

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
