## [Decision Letter · Decision Letter 0]

8 Feb 2024

PONE-D-23-25640Introducing heat-not-burn tobacco improves hematocrit and cigarette smoking-related symptoms in patients with smokers’ polycythemia and polycythemia veraPLOS ONE

Dear Dr. iizuka,

Thank you for submitting your manuscript to PLOS ONE. After careful consideration, we feel that it has merit but does not fully meet PLOS ONE’s publication criteria as it currently stands. Therefore, we invite you to submit a revised version of the manuscript that addresses the points raised during the review process.

We look forward to receiving your revised manuscript.

Kind regards,

Billy Morara Tsima, MD MSc

Academic Editor

PLOS ONE

Journal Requirements:

When submitting your revision, we need you to address these additional requirements. 1. Please ensure that your manuscript meets PLOS ONE's style requirements, including those for file naming. The PLOS ONE style templates can be found at https://journals.plos.org/plosone/s/file?id=wjVg/PLOSOne_formatting_sample_main_body.pdf and https://journals.plos.org/plosone/s/file?id=ba62/PLOSOne_formatting_sample_title_authors_affiliations.pdf 2. Note from Emily Chenette, Editor in Chief of PLOS ONE, and Iain Hrynaszkiewicz, Director of Open Research Solutions at PLOS: Did you know that depositing data in a repository is associated with up to a 25% citation advantage (https://doi.org/10.1371/journal.pone.0230416)? If you’ve not already done so, consider depositing your raw data in a repository to ensure your work is read, appreciated and cited by the largest possible audience. You’ll also earn an Accessible Data icon on your published paper if you deposit your data in any participating repository (https://plos.org/open-science/open-data/#accessible-data). 3. In the online submission form, you indicated that " The datasets generated and/or analyzed during the current study are not publicly available because many cases are also used for research at each facility; however, they are available from the corresponding author upon reasonable request." All PLOS journals now require all data underlying the findings described in their manuscript to be freely available to other researchers, either 1. In a public repository, 2. Within the manuscript itself, or 3. Uploaded as supplementary information.This policy applies to all data except where public deposition would breach compliance with the protocol approved by your research ethics board. If your data cannot be made publicly available for ethical or legal reasons (e.g., public availability would compromise patient privacy), please explain your reasons on resubmission and your exemption request will be escalated for approval. 

Reviewers' comments:

Reviewer's Responses to Questions

**Comments to the Author**

1. Is the manuscript technically sound, and do the data support the conclusions?

Reviewer #1: Partly

Reviewer #2: Partly

2. Has the statistical analysis been performed appropriately and rigorously? 

Reviewer #1: I Don't Know

Reviewer #2: Yes

3. Have the authors made all data underlying the findings in their manuscript fully available?

Reviewer #1: Yes

Reviewer #2: Yes

4. Is the manuscript presented in an intelligible fashion and written in standard English?

Reviewer #1: Yes

Reviewer #2: No

5. Review Comments to the Author

Reviewer #1: Polycythemia in smokers is common in hematology practice, and they often have difficulty quitting smoking. This study is novel as the first relevant study to examine whether switching from cigarettes to heat-not-burn (HNB) tobacco improves polycythemia and associated symptoms.

I believe that the results of this study will stimulate other similar studies in many areas with large sample sizes to study the improvement of erythrocytosis by smoking cessation through heatless tobacco (HNB) and also e-vaping, in patients with polycythemia in smokers. I highly recommend publishing it.

Here just some comments prefer to clear:

- Did smokers polycethemia patient - besides the heat-not-burn (HNB) tobacco- undergo therapeutic venesection (phlebotomy)?? ((- as this procedure widely and commonly used to decrease hematocrit in smokers polycethemia to reduce hyperviscosity and consequently reduce incidence of thrombosis))

- In method, some details should be mention like the study area (high attitude….), medical history of patients (obesity, cardiac disease, obstructive sleep apnea ), Also definition and details regarding (HNB) tobacco using.

- In method, patient, line -6 -((.. From August 1, 2009 to December 31, 2021, smokers who met the WHO 2016 diagnostic criteria for hematocrit (Hct) and Hb of polycythemia vera (Hct >49% and/or Hb 16.5 g/dL in males; Hct >48% and/or Hb 16.0 g/dL in females) who had failed a challenge of smoking cessation were recruited The patient differentiation process is shown in Figure 1)). Please recheck sentence as according figure 1 a fifteen where recruited, They quit smoking, and switch to HNB tobacco (NOT fail a challenge of smoking cessation.

- The study was from August 1, 2009 to December 31, 2021, that’s mean the study period was 11 years. How many patients of smokers polycethemia included in that period ? Can you compare hematocrit between smokers with polycythemia who failed the smoking cessation challenge and others who chose to quit smoking, or switch to tobacco HNB. Clarifying whether smokers with polycythemia who fail a smoking cessation challenge receive treatment or therapeutic phlebotomy. I believe that this clarification is needed in the study because the study period was long and the sample enrolled was small.

Reviewer #2: Please see the uploaded document with all the reviewer comments for more detail.

The findings of this study is interesting and should be shared.

The ideas behind the manuscript are clear, however, some of the concepts and sentence structures may have been lost in translation.

6. PLOS authors have the option to publish the peer review history of their article (what does this mean? ). If published, this will include your full peer review and any attached files.

**Do you want your identity to be public for this peer review?** For information about this choice, including consent withdrawal, please see our Privacy Policy .

Reviewer #1: No

Reviewer #2: **Yes: ** Ethan J Gantana

---

## [Author Response · Author response to Decision Letter 1]

19 May 2024

Response to Reviewer: 1

We wish to express our appreciation to the reviewer for his or her insightful comments, which have helped us improve the paper.

- Did smokers polycethemia patient - besides the heat-not-burn (HNB) tobacco- undergo therapeutic venesection (phlebotomy)?? ((- as this procedure widely and commonly used to decrease hematocrit in smokers polycethemia to reduce hyperviscosity and consequently reduce incidence of thrombosis))

Response: Thank you for pointing this out. Two patients had undergone phlebotomy (polycythemia vera; n=1, smokers’ polycythemia; n=1), but they had stopped phlebotomy more than three months before the start of the clinical trial. Phlebotomy was not resumed during the study period.

The following explanation has been added to the "Treatment administration and assessment of outcomes" in the Methods section.

→

“Two patients had undergone phlebotomy but had discontinued phlebotomy more than 3 months before entering this study, and there was no resumption of phlebotomy during the observation period.”

- In method, some details should be mention like the study area (high attitude….), medical history of patients (obesity, cardiac disease, obstructive sleep apnea), Also definition and details regarding (HNB) tobacco using.

Response: Thank you for pointing this out. The following explanation has been added to the "Diagnosis" of the Methods section. We described the fact that we ruled out any medical history that could be a cause of secondary polycythemia other than smoking and added the method for diagnosing smoking polycythemia. In addition, information about altitude has been added to the "Patient" section of the results.

→

“Diagnosis

The patient differentiation process is shown in Figure 1. In smokers with polycythemia, causes of secondary polycythemia other than smoking were excluded: obesity (BMI ≥ 35), diabetes mellitus, psychological stress, sleep apnea syndrome, hyperthyroidism (hypertensive patients only). In addition, it was confirmed that erythropoietin was normal or low, and BCR-ABL gene fusion and JAK2 mutations (V617F and exon 12 mutation) were negative. Furthermore, patients who were negative for JAK2 mutations, whose bone marrow biopsy did not show hypercellularity with trilineage growth, were diagnosed as having smokers’ polycythemia.”

Results

Patients

“Of the 40 patients with polycythemia who were smokers, the cause of polycythemia was investigated and differentiated in 26 patients. They lived between 20 and 70 m above sea level.”

- In method, patient, line -6 -((.. From August 1, 2009 to December 31, 2021, smokers who met the WHO 2016 diagnostic criteria for hematocrit (Hct) and Hb of polycythemia vera (Hct >49% and/or Hb 16.5 g/dL in males; Hct >48% and/or Hb 16.0 g/dL in females) who had failed a challenge of smoking cessation were recruited The patient differentiation process is shown in Figure 1)). Please recheck sentence as according figure 1 a fifteen where recruited, They quit smoking, and switch to HNB tobacco (NOT fail a challenge of smoking cessation.

Thank you for pointing this out. Figure 1 and the text have been modified.

→

“From August 1, 2009, to December 31, 2021, we recruited smokers who met the WHO 2016 hematocrit (Hct) and Hb diagnostic criteria [10] for polycythemia vera (Hct >49% and/or Hb 16.5 g/dL in males; Hct >48% and/or Hb 16.0 g/dL in females) and who had a history of failing a challenge of smoking cessation in the past.”

- The study was from August 1, 2009 to December 31, 2021, that’s mean the study period was 11 years. How many patients of smokers polycethemia included in that period ? Can you compare hematocrit between smokers with polycythemia who failed the smoking cessation challenge and others who chose to quit smoking, or switch to tobacco HNB. Clarifying whether smokers with polycythemia who fail a smoking cessation challenge receive treatment or therapeutic phlebotomy. I believe that this clarification is needed in the study because the study period was long and the sample enrolled was small.

Response:

Thank you very much for your excellent suggestion.

We identified 40 smokers with polycythemia, and 26 patients were examined for other causes of polycythemia. Seven of the 26 patients were excluded owing to obesity (n=2), diabetes mellitus (n=4), or mental stress at work (n=1). Two of the 19 patients were JAK2V617F positive and were diagnosed with polycythemia vera. Seventeen were diagnosed with smokers’ polycythemia, but four chose to continue smoking. Thirteen patients with smokers' polycythemia and two patients with polycythemia vera quit smoking or switched to HNB tobacco. We have added these explanations to the “Patients” section in the Results.

No patients resumed smoking cigarettes after switching to HNB tobacco. However, two of the four patients who chose to quit smoking resumed their smoking habits. We have added this information to the “Long-term changes in the smoking habit” section of the Results.

→

Results

Patients

“Of the 40 patients with polycythemia who were smokers, the cause of polycythemia was investigated and differentiated in 26 patients. They lived between 20 and 70 m above sea level. Seven patients were excluded owing to obesity (BMI>35; n=2), diabetes mellitus (n=4), or mental stress at work (n=1). Two of the 19 patients were JAK2V617F positive and were diagnosed with polycythemia vera. Seventeen patients were diagnosed with smokers’ polycythemia, but four did not participate in the study because they continued smoking (Figure 1). Therefore, 13 patients with smokers’ polycythemia and two patients with JAK2V617F-positive polycythemia vera were enrolled in this study (Figure 1).”

Reasons why the number of registered patients was small compared to the registration period

1. Only patients with a history of failed smoking cessation challenges in the past were recruited.

2. There was a lack of tests to determine the differential diagnosis, which limited the number of eligible patients.

3. Physicians other than hematologists were not yet accurately aware of the complications (eg, thrombosis) risks of secondary polycythemia, and referrals to the hematology department were rare.

4. Owing to institutional ethical standards, only patients referred to the Department of Hematology were allowed to participate in the study.

It was not possible to include in the main text the follow-up of four smoking polycythemia patients who were not enrolled in the study (continued to smoke). However, for patients whose smoking amount returned to pre-study levels, Hct level returned to the same level at the enlloled of this study. We have added this fact to the end part of "Long-term changes in smoking habits" section.

→

Long-term changes in the smoking habit

“Patients who returned cigarette consumption to before smoking cessation, their Hct level returned to the same level at the enrollment in this study (data not shown).”

As reference, we write down the changes of Hct and WBC in four smokers’ polycythemia patients who continued to smoking.

→

“No change was observed in the WBC or Hct of the four patients who continued smoking (WBC, 9150±1277/µL vs. 10650±1546/µL, p=0.400; Hct, 53.8±3.26% vs. 55.75±2.36, p=0.629).”

Response to Reviewer: 2

We wish to express our appreciation to the reviewer for his or her insightful comments, which have helped us improve the paper.

-The ideas behind the manuscript are clear, however, some of the concepts and sentence structures may have been lost in translation.

Response:

Thank you for pointing this out. We reviewed the text and added or modified missing explanations. Our manuscript has undergone professional English editing to ensure correct grammar and a native English tone.

Abstract

• Objectives as a first heading should be changed to Introduction instead.

• Remove “on”.

• Methods section in abstract requires refinement.

Response:

Thank you for pointing this out. We deleted the objectives from the abstract and included this information in the introduction. In addition, we removed “on.”

Following PLOS One's style also solved the problem in the Methods section.

Introduction

• A reference at the end of the sentence would suffice and prevent the flow of the sentence from being interrupted.

• Also consider HIF as a mechanism for absolute erythrocytosis (esp in patients with COPD)

o However, the authors should mention and refer to reputable sources that EPO is low in smokers, in contrast to other causes of secondary polycythaemia where an elevated serum EPO level is to be expected.

Response:

Thank you for pointing this out. We have added a statement that smokers tend to have polycythemia even with low EPO levels, and we added one more article to the references.

Grammar and sentence structures should be checked to ensure that the correct words and terms match the meaning/intention of the sentences. This seems to have been lost in the translation of the manuscript.

Response:

Thank you for pointing this out. The references have been changed to only be listed at the end of the text. However, some references were left in the middle of the text to make it easier for readers to search.

Methods:

• Please write genes and fusion genes such as “BCR::ABL1” in Italics instead of “BCR-ABL”

Response: Thank you for pointing this out. I have corrected it as you pointed out.

• FISH sensitivity to detect BCR::ABL1 is very low; why was PCR not used and were there any clinicopathological features of CML in any of these patients?

Response:

To differentiate polycythemia vera, it is necessary to confirm that BCR-ABL is negative (WHO 2017 diagnostic criteria). Therefore, BCR-ABL was measured even in the absence of typical clinicopathological features of CML.

The accuracy of FISH and the reasons for conducting it are as follows.

1. When we contacted SRL, we found that FISH had the same detection level as 10*2 copies of PCR.

2. The FISH diagnostic testing method is recommended, like PCR, in the NCCN (National Comprehensive Cancer Network) guidelines and the Japanese Guidelines.

3. There are very few facilities that can measure micro BCR-ABL using the PCR method. Japan Health Insurance requires the PCR method to measure major BCR-ABL and minor BCR-ABL in different months. In addition, measuring micro BCR-ABL requires more time.

Since the purpose of this study is actual medical treatment, we proposed a method that is in line with these guidelines and actual clinical practice. We also believe that simpler methods are more versatile worldwide.

• Define “SRL” before using it as an abbreviation.

Response: Thank you for pointing this out. I checked with the company, but SRL is the official name (not an acronym or an abbreviation). We have changed the name to SRL, Inc., to avoid misunderstanding.

• Provide more detail in the “Gene analysis” section.

o Was a positive/mutant (heterozygous/homozygous), a negative/wild-type and a reagent blank control used during the JAK2 mutation test?

o Please state whether the PCR products (end-point analysis) were read on gel cards.

Response: I checked with the company. AS (Allele Specific) PCR uses the same reverse primer and two different forward primers for the normal allele and the mutant allele to compete in the PCR reaction to semi-quantitatively measure the JAK2 mutation, which is a single nucleotide mutation. Therefore, it is possible to quantify Allele burden. Real-time PCR uses Qiagen's rotor G, so no gel card is used.

• Define what exactly is meant by ‘heat-not-burn’ (HNB) tobacco in the method section under “Treatment administration and assessment of outcome”.

Response: Thank you for pointing this out. Added that HNB tobacco used Ploom S, Ploom X, and IQOS ILUMA.

Was the serum EPO level recorded, if not, why not?

Response: EPO levels are listed in Supplementary Table 1. A comparison between the HNB tobacco group and the non-smoking group is also listed in Table 1. Because this study focused on changes in Hct and clinical symptoms, EPO was measured only at the first visit. Additionally, frequent EPO measurements were not performed in this study because they are not permitted by Japanese guidelines or the insurance system.

• Which programs/software were used for the statistical analysis and graphical representation of the data?

Response: We used R soft. We added this information to the Methods section.

• How was the sputum volume objectively measured or is this also a subjective assumption?

Response: It was confirmed that the number of times spittoon per day was 0. Therefore, there is no doubt that their sputum completely disappeared after switching to HNB cigarettes, but we did not measure the amount.

• Hct is the only objective measure of improvement.

o Was % blood oxygen saturation assessed for each of the patients at baseline and follow-up?

Response: Blood oxygen saturation was not measured. This is because oxygen saturation has not been shown to be involved in thrombosis or mortality in secondary polycythemia, including smoking polycythemia. The guidelines do not include SPO2 as an observation item, and Hct is the main indicator for treatment. However, with regard to respiratory status, all patients were grade I on the Fletcher-Hugh-Jones classification and grade 0 on the MRC (Medical Research Council) dyspnea scale.

・ The authors mention a mechanism of reduced plasma cell volume with cigarette smoking. Was systemic blood pressure affected by the switch to HNB or smoking cessation?

Response; Plasma cells differentiated from B lymphocytes are not described. Plasma volume is not measured or calculated because it is difficult to measure directly, and it is easily affected by eating, drinking, and other physical conditions. Blood pressure was not measured because the body adjusts it to some extent even if there is a change in circulating plasma volume. However, systolic blood pressure was measured at every consultation, and it was 100 before switching vs. after 200 switching (p=0.81).

• Were baseline lung function tests performed?

Response; Thank you for pointing this out. We did not perform lung function tests. However, all patients were grade I on the Fletcher-Hugh-Jones classification and grade 0 on the MRC (Medical Research Council) dyspnea scale. In addition, no patients were diagnosed with COPD on chest X-ray. We have added this information to the Patients section of the Methods.

• Is there a follow-up planned for incidence of thromboembolism in these patients?

Response: We are in the process of increasing the number of participating hospitals (currently under review by the Ethics Committee), and we plan to add not only thrombosis but also larger-scale and more detailed analyses (COHb measurements, secondary cancer rates, etc.).

Results

Table 1 – Total number is 12 but HNB is 8 and smoking cessation is 5 which equals 13. Was the one patient that had an unsuccessful smoking cessation, excluded from the data? I agree with the exclusion, but the authors need to state this exclusion and change the values in the table.

Response: Thank you for pointing this out. The total number is 13. The total number in Table1 was incorrect. We corrected it.

Discussion

“One possible mechanism is that absolute erythrocytosis which activates thrombin21 and increases platelet P secretin expression,22 which in turn increases the risk of thrombosis.” Did the authors mean “P selectin”?

Response: Thank you for pointing this out. We corrected it as P selectin.

“As the risk of secondary carcinogenesis related to cigarettes, cigarette smoking increases the incidence of polycythemia vera,26,27 as well as 16 types of cancer: lung, upper aerodigestive tract (oral cavity, nasal cavity, nasal sinus, pharynx, larynx), esophagus, stomach, pancreas, liver, kidney, lower urinary tract (renal pelvis and urinary bladder), and uterine cervix cancers, myelogenous leukemia, and myeloproliferative neoplasms. 28” This sentence seems incomplete.

Response: Thank you for pointing this out. The text has been modified.

→

Regarding the risk of secondary carcinogenesis related to cigarettes, cigarette smoking increases the incidence of 17 types of cancer: lung, upper aerodigestive tract (oral cavity, nasal cavity, nasal sinus, pha

---

## [Decision Letter · Decision Letter 1]

19 Jun 2024

PONE-D-23-25640R1Introducing heat-not-burn tobacco improves hematocrit and cigarette smoking-related symptoms in patients with smokers’ polycythemia and polycythemia veraPLOS ONE

Dear Dr. iizuka,

Thank you for submitting your manuscript to PLOS ONE. After careful consideration, we feel that it has merit but does not fully meet PLOS ONE’s publication criteria as it currently stands. Therefore, we invite you to submit a revised version of the manuscript that addresses the points raised during the review process.

We look forward to receiving your revised manuscript.

Kind regards,

Billy Morara Tsima, MD MSc

Academic Editor

PLOS ONE

Reviewers' comments:

Reviewer's Responses to Questions

**Comments to the Author**

1. If the authors have adequately addressed your comments raised in a previous round of review and you feel that this manuscript is now acceptable for publication, you may indicate that here to bypass the “Comments to the Author” section, enter your conflict of interest statement in the “Confidential to Editor” section, and submit your "Accept" recommendation.

Reviewer #1: (No Response)

Reviewer #2: (No Response)

2. Is the manuscript technically sound, and do the data support the conclusions?

Reviewer #1: Partly

Reviewer #2: Partly

3. Has the statistical analysis been performed appropriately and rigorously? 

Reviewer #1: Yes

Reviewer #2: Yes

4. Have the authors made all data underlying the findings in their manuscript fully available?

Reviewer #1: Yes

Reviewer #2: Yes

5. Is the manuscript presented in an intelligible fashion and written in standard English?

Reviewer #1: Yes

Reviewer #2: Yes

6. Review Comments to the Author

Reviewer #1: This manuscript is very interesting, as the number of smokers is increasing among the population around the world and most of them are males and may be most of them are heavy smoker too. Smokers are exposed to many toxic substances. Although there are no accurate statistics, but it is expected that the number of smoker polycythemia will increase significantly resulting a known serious complication.

The importance of this study is, it is a first relevant study to examine whether switching from cigarettes to heat-not-burn (HNB) tobacco improves polycythemia and associated symptoms.

A drawback of this study is that it has several limitations, one of which is the small sample size despite the long study period (although the author explains the reasons to the reviewer not in the manuscript.

The study remains publishable and an incentive to conduct similar studies while overcoming these limitations.

I hope the author adds the limitations and the recommendations in the manuscript , Since the author has a plan to increase the number of participating hospitals with larger-scale and more detailed analyses, so the results then will be enhanced and improved to be to be widely applicable

Reviewer #2: Not all of the conclusions or statements made in the manuscript about HNB tobacco can be supported by the data presented. Please see the attached document for all comments with regard to revisions.

7. PLOS authors have the option to publish the peer review history of their article (what does this mean? ). If published, this will include your full peer review and any attached files.

**Do you want your identity to be public for this peer review?** For information about this choice, including consent withdrawal, please see our Privacy Policy .

Reviewer #1: **Yes: ** Anisa H. Albiti

Reviewer #2: **Yes: ** Ethan Gantana

---

## [Author Response · Author response to Decision Letter 2]

30 Sep 2024

We wish to express our appreciation to the reviewers again for his or her comments, which have helped us improve the paper.

Could the authors of this study please provide a conflict-of-interest statement.

→In addition to the COI, we have also added Author contributions and a Data Availability Statement.

HUGO Gene Nomenclature Committee (HGNC) recommendations for the designation of gene fusions, recommends writing a fusion gene as BCR::ABL1.

• Bruford, E.A., Antonescu, C.R., Carroll, A.J. et al. HUGO Gene Nomenclature Committee (HGNC) recommendations for the designation of gene fusions. Leukemia 35, 3040–3043 (2021). https://doi.org/10.1038/s41375-021-01436-6

• Also see Nomenclature under the Submission guidelines for PLOS ONE (https://journals.plos.org/plosone/s/submission-guidelines)

The term ‘variant’ should be used instead of ‘mutation’ throughout the manuscript.

Subject to the requirements of PLOS ONE, it should be sufficient to place a reference at the end of the sentence and to improve the "flow" of the sentence.

→I have corrected it as you pointed out.

Abstract

Line 35: write 2 as “two” instead.

→I have corrected it as you pointed out.

Introduction

Line 52: mechanism or plural mechanisms? Change the grammar accordingly.

→It is plural because it refers to the mechanisms in lines 48-52.

Line 58: Treatment for smokers’ polycythaemia?

→ Generally, patients with smoker’s polycythemia are not given medical treatment. However, guidelines indicate phlebotomy if the Hct is >55% and/or the patient is symptomatic. In this study, as described in the "Treatment administration and assessment of outcomes" section of the Methods, two patients had undergone phlebotomy, one of whom had been diagnosed with smoker’s polycythemia. However, as described in “Treatment administration and assessment of outcomes” section of the manuscript, all patients had discontinued phlebotomy >3 months before entering this study. Therefore, patients with smoker’s polycythemia were not receiving treatment at study initiation.

Line 59: suggest using “detection or analysis or testing” rather than “measurement”

→I have corrected it as you pointed out.

Line 60-61: Do the authors mean that the distinction/differentiation between a diagnosis of PV and secondary polycythaemia has become more accurate?

→Thanks, you are right.

Line 59-63: This needs clarification. Do the authors mean that because we are now able to better distinguish between smokers’ (i.e secondary) polycythaemia and polycythaemia vera by using JAK2 variant analysis, that the results of the study in reference 10, in which JAK2 testing was performed, is more accurate than the 1993 report (case-control study), in determining which condition is more associated with thrombosis?

If this is the case, I suggest that the authors instead mention that the fact that JAK2 variants were not tested for is a limitation of the 1993 report and not the reason why the results of reference 10 suggest an equivalent thrombosis rate. Example: “In contrast to these earlier reports, where PV was not confirmed by testing for the JAK2 variant, a more recent study has shown that secondary polycythaemia and polycythaemia vera have comparable rates of thromboembolic events. This suggests that secondary polycythaemia also requires intervention to reduce the risk of thrombosis in this cohort.”

→I have corrected the sentence as per your advice.

[The authors should also consider this article (Nadeem O, Gui J, Ornstein DL. Prevalence of venous thromboembolism in patients with secondary polycythemia. Clin Appl Thromb Hemost. 2013;19(4):363-366. doi:10.1177/1076029612460425) which states that “…secondary polycythemia alone may not be a significant risk factor for VTE but that VTE risk in this population may be related to known risk factors such as obesity.”]

→I am aware of this study. However, it focuses only on venous thrombosis. As described in references 10, 26, and 27, both secondary polycythemia and PV have higher risks of thrombosis in the arteries than in the veins. In addition, the study you mentioned included patients with severe obesity (Max BMI of 59.5 kg/m2). In contrast, the present study excluded participants with severe obesity (the maximum BMI in this study was 33.51 kg/m2). Generally, a BMI >30 kg/m2 is associated with an increased risk of venous thrombosis due to obesity. However, all but one patient had a BMI <30 kg/m2. Therefore, I believe that the study you mentioned is not relevant to the present study and should not be cited.

Line 68: I suggest: “…increased the need for consultation with a hematologist.”

→I have corrected the sentence as per your advice.

Line 75-77: The authors are making a bold statement about HNB tobacco as a ‘therapeutic aid’. Please rephrase this statement.

→The text has been corrected to read as follows.

“we report that switching to HNB tobacco improved smokers’ polycythemia and subjective symptoms with smoking in smokers. ”

Method

Line 112-113: Add the abbreviation after polymerase chain reaction (PCR), which is used later in the paragraph, and move the AS abbreviation to allele-specific instead of putting it after 'polymerase'.

→Thank you for your advice. I Corrected fixed it.

Please define FISH in the manuscript before using the abbreviation and provide sufficient methodological details in the manuscript for “FISH at SRL, Inc” and the PCR used for JAK2 testing.

→ “FISH” has been corrected to "fluorescence in-situ hybridization." Regarding the PCR, any information beyond that reflected in the response to the previous peer reviewer is confidential and cannot be provided by SRL, Inc. Therefore, I apologize that I cannot provide additional details.

Was the 2016 WHO classification also used for the diagnosis of patients presenting before the publication of this classification?

→Absolutely. All patients were re-diagnosed when the paper was written to ensure uniformity of diagnosis by WHO2016.

The authors do not need to add trade/brand names or give examples of HNB products. However, they must define what HNB is and how it differs from cigarette smoking.

→Thank you for pointed out.

The patient section was revised to clarify that the recruited participants were cigarette smokers with polycythemia. I hope that this makes it clear from the flow of the text that smoking means cigarette smoking, while HNB tobacco refers to heat-not-burn tobacco.

Results

Lines 138-147 are perhaps better suited for the method section. These are not senso stricto ‘results’.

→I'm sorry. For the sake of the flow of the article, I will just put it in the results section.

Line 196-198: Please revise the sentence for clarification. Do the authors mean to say “In patients who resumed cigarette consumption after smoking cessation, their Hct levels returned to the same level as at enrolment in this study?

• This statement is irrelevant if there is no data.

→I have the data. I added it to Manuscript.

Discussion

Line 228: “Therefore, smoking cessation is more urgent for those with polycythemia vera compared with those with smokers’ polycythemia.” This stands in contrast to what the authors are trying to say in the introduction lines 62-63. Why is it more urgent for those with PV when the authors discuss in the introduction that PV and secondary erythrocytosis have equivalent rates of thrombosis?

I apologize for the confusion. This is not contrast.

The incidence of thrombosis in patients with polycythemia vera (non-smoker PV) is the same as that in patients with smoker's polycythemia. However, it is well known that the incidence of arterial thrombosis in patients with polycythemia vera who smoke is 1.9 times higher than in patients who do not smoke. Thus, among patients with polycythemia vera, smokers have a higher risk of thrombosis than non-smokers.

To clarify this point, I have revised the sentence.

→The incidence of arterial thrombosis in patients with polycythemia vera who smoke is 1.9 times higher than in patients who are non-smokers [26, 27], and whose high Hct levels also increase the risk of thrombosis [28]. In this study, it was intriguing that in two polycythemia vera patients who have dropped out of smoking cessation programs, high Hct levels improved after switching to HNB tobacco without additional treatment.

Line 231: The results of this study ONLY SUGGESTS that HNB tobacco MAY be useful in decreasing the Hct and smoking-related symptoms in patients with secondary polycythaemia. No claims can be made with regards to PV and certainly not with regards to using HNB tobacco as “an emergency measure in patients with polycythemia vera…” to reduce the Hct. This claim is too bold with only 2 patients in the PV arm and no sufficient scientific evidence to make this statement. I ask that the authors revise this statement. The only evidence from this particulate study is that 2 patients with confirmed PV who switched to HNB tobacco had lower Hct levels after the switch.

→No other reports have described a decrease in Hct levels in patients with confirmed PV who switched to HNB cigarettes. Although the present study included two cases, this fact alone is important, so I have revised the sentence to remove any exaggerated claims.

Lines 232 – 239 are irrelevant to the study. The only qualification the authors have to make is that HNB, like normal cigarettes, contains harmful substances and the long-term effects have not yet been clarified.

The long-term effects are not described. Ten years have passed since HNB cigarettes were released, and new findings have emerged; therefore, the newer findings may differ from those of previous reports.

However, the following two points have been reported:

- The smoke of HNB cigarettes contains fewer harmful substances compared with paper cigarettes.

-Compared to smokers of paper cigarette, smokers of HNB cigarettes exhibit less DNA methylation of certain cancer-related genes.

The present report indicates that the health effects of HNB cigarettes remain unclear. We believe that the more information we provide our readers about HNB tobacco, the better, as a criterion for its application in clinical practice. Although this is often the subject of debate, there is no intention to mislead readers.

Line 242: Again. Although the results of this study are interesting, especially with regard to secondary polycythaemia, this study ONLY SUGGESTS that Hct MAY be reduced by HNB tobacco use compared to cigarette smoking in patients with secondary erythrocytosis. No conclusions can be drawn about PV or thrombosis risk. Further studies are needed to assess a reduction in the risk of thrombosis.

→I corrected this sentence to say only that it improved Hct.

→In this study, it was intriguing that in two polycythemia vera patients who have dropped out of smoking cessation programs, high Hct levels improved after switching to HNB tobacco without additional treatment.

I disagree with the authors that HNB tobacco “may become a therapeutic aid” (line 76) or is considered a “treatment option” (line 247). This could be misleading to many readers of this article. At best, HNB tobacco should be considered as an alternative to cessation of cigarette smoking when cessation of cigarette smoking is unsuccessful.

→I have revised this sentence as follows. However, I do not believe the original text is mistaken considering the trends in recent research reports.

Line 76: Herein, we report that switching to HNB tobacco improves polycythemia and subjective symptoms in smokers.

Line 247: Although complete smoking cessation is the best treatment, HNB tobacco may be considered as an alternative when cigarette smoking cessation is unsuccessful.

Figure 1:

• “Differential diagnosis not performed” as a reason for excluding 14 patients. What does this mean? The reason(s) for their exclusion is not clear.

→These are patients for whom consent was obtained, but for whom the attending physician had not performed sufficient tests for differential diagnosis. Some of the tests to check for obesity, hyperthyroidism, diabetes, or high Epo levels were not done. In addition, the medical interview did not mention whether the patient was experiencing mental stress.

• What is being referred to with “mental storess at work”? Does this refer to how psychological stress was assessed? How was this determined to sufficiently exclude the patient from the study?

→The patient was under considerable stress at work, and his polycythemia improved when he was transferred to another department. This occurred when multiple blood samples were collected before switching to HNB.

---

## [Decision Letter · Decision Letter 2]

1 Apr 2025

PONE-D-23-25640R2Introducing heat-not-burn tobacco improves hematocrit and cigarette smoking-related symptoms in patients with smokers’ polycythemia and polycythemia veraPLOS ONE

Dear Dr. Iizuka,

Thank you for submitting your manuscript to PLOS ONE. After careful consideration, we feel that it has merit but does not fully meet PLOS ONE’s publication criteria as it currently stands. Therefore, we invite you to submit a revised version of the manuscript that addresses the points raised during the review process.

We look forward to receiving your revised manuscript.

Kind regards,

Billy Morara Tsima, MD MSc

Academic Editor

PLOS ONE

Journal Requirements:

Reviewers' comments:

Reviewer's Responses to Questions

**Comments to the Author**

1. If the authors have adequately addressed your comments raised in a previous round of review and you feel that this manuscript is now acceptable for publication, you may indicate that here to bypass the “Comments to the Author” section, enter your conflict of interest statement in the “Confidential to Editor” section, and submit your "Accept" recommendation.

Reviewer #1: (No Response)

Reviewer #2: All comments have been addressed

2. Is the manuscript technically sound, and do the data support the conclusions?

Reviewer #1: Partly

Reviewer #2: Yes

3. Has the statistical analysis been performed appropriately and rigorously? 

Reviewer #1: N/A

Reviewer #2: Yes

4. Have the authors made all data underlying the findings in their manuscript fully available?

Reviewer #1: Yes

Reviewer #2: Yes

5. Is the manuscript presented in an intelligible fashion and written in standard English?

Reviewer #1: Yes

Reviewer #2: Yes

6. Review Comments to the Author

Reviewer #1: Despite the study limitations, this manuscript is interesting and will be useful. I emphasize that the author should add limitations and recommendations

the results of this study suggest well that HNB tobacco may be so helpful in smoking-related symptoms in patients secondary polycythemia. But for smoker polycythemia vera patients, although the result in study is intriguing, it is also metaphorical. To reach the intended conclusion, the author would need to compare results between groups of smoker with polycythemia vera patient who are on treatment ( who have fail to quit smoking and who switch to HNB tobacco), mentioning limitations here is demanding

Reviewer #2: The article has been considerably improved since the first revision. At this stage, only minor revisions are necessary.

Minor revisions (line citation based on marked-up document):

Introduction

Line 56: The authors have changed the text here, so abbreviations must first be defined before they are used e.g. PV before using it for the first time and when it is defined, the authors do not need to write it in full (e.g. line 58). Please see the rest of the text for similar instances.

Results:

Lines 186-189: This seems to be a mistake. Repetition of the previous lines.

Comments from previous review:

Reviewer comment: The authors do not need to add trade/brand names or give examples of HNB products. However, they must define what HNB is and how it differs from cigarette smoking.

Author response: →Thank you for pointed out. The patient section was revised to clarify that the recruited participants were cigarette smokers with polycythemia

• The authors have not clearly defined HNB or explained how it differs from traditional cigarette smoke, assuming prior reader knowledge. Simply listing brand names is insufficient; a brief comparison should be included in my opinion. Important distinctions such as combustion in cigarettes (high temperature, smoke and by-products) versus heating in HNB (lower temperature, no combustion, aerosol production) would improve clarity. If the editor(s) feel that this level of detail is unnecessary, I am open to the authors not providing any additional response. Of note, the differences in CO content are addressed in the discussion.

Reviewer comment: Lines 138-147 are perhaps better suited for the method section. These are not senso stricto ‘results’.

Author response: →I'm sorry. For the sake of the flow of the article, I will just put it in the results section.

• I still disagree that this needs to be in the results section (the first paragraph of the results section under “Patients"), but if the editor(s) are happy with the placement of the text on metholodology (describing the inclusion and exclusion of patients) in the results section, then I will overlook this response. In my opinion, adding this information to the methods section would improve "flow of the article”.

7. PLOS authors have the option to publish the peer review history of their article (what does this mean? ). If published, this will include your full peer review and any attached files.

**Do you want your identity to be public for this peer review?** For information about this choice, including consent withdrawal, please see our Privacy Policy .

Reviewer #1: **Yes: ** Anisa Albiti

Reviewer #2: **Yes: ** Ethan James Gantana

---

## [Author Response · Author response to Decision Letter 3]

6 Apr 2025

We wish to express our appreciation to the Reviewer for his or her insightful comments, which have helped us significantly improve the paper.

Reviewer #1: Despite the study limitations, this manuscript is interesting and will be useful. I emphasize that the author should add limitations and recommendations

the results of this study suggest well that HNB tobacco may be so helpful in smoking-related symptoms in patients secondary polycythemia. But for smoker polycythemia vera patients, although the result in study is intriguing, it is also metaphorical. To reach the intended conclusion, the author would need to compare results between groups of smoker with polycythemia vera patient who are on treatment (who have fail to quit smoking and who switch to HNB tobacco), mentioning limitations here is demanding

Answer: It is corrected that by adding "However, due to the small sample size, further expanded studies are needed." to the "Discussion" section, we avoid misleading the readers and indicate that this is a topic for future research.

Reviewer #2: The article has been considerably improved since the first revision. At this stage, only minor revisions are necessary.

Minor revisions (line citation based on marked-up document):

Introduction

Line 56: The authors have changed the text here, so abbreviations must first be defined before they are used e.g. PV before using it for the first time and when it is defined, the authors do not need to write it in full (e.g. line 58). Please see the rest of the text for similar instances.

Answer: Thank you for pointing this out. We have corrected it as you pointed out.

Results:

Lines 186-189: This seems to be a mistake. Repetition of the previous lines.

Answer: Thank you for pointing this out. We have deleted the phrase.

Comments from previous review:

Reviewer comment:

The authors do not need to add trade/brand names or give examples of HNB products. However, they must define what HNB is and how it differs from cigarette smoking.

• The authors have not clearly defined HNB or explained how it differs from traditional cigarette smoke, assuming prior reader knowledge. Simply listing brand names is insufficient; a brief comparison should be included in my opinion. Important distinctions such as combustion in cigarettes (high temperature, smoke and by-products) versus heating in HNB (lower temperature, no combustion, aerosol production) would improve clarity. If the editor(s) feel that this level of detail is unnecessary, I am open to the authors not providing any additional response. Of note, the differences in CO content are addressed in the discussion.

Answer: We have reconsidered and added the following text to the “Introduction” section. We also removed the brand name.

“The characteristic of HNB tobacco is no combustion, lower temperature than cigarettes and do not produce smoke containing CO. On the other hand, metal aerosols are generated.”

• I still disagree that this needs to be in the results section (the first paragraph of the results section under “Patients"), but if the editor(s) are happy with the placement of the text on metholodology (describing the inclusion and exclusion of patients) in the results section, then I will overlook this response. In my opinion, adding this information to the methods section would improve "flow of the article”.

Answer: We have reconsidered and added it to the “Methods” section as your suggested.

---

## [Editor Report · Decision Letter 3]

9 Apr 2025

Introducing heat-not-burn tobacco improves hematocrit and cigarette smoking-related symptoms in patients with smokers’ polycythemia and polycythemia vera

PONE-D-23-25640R3

Dear Dr. Iizuka,

We’re pleased to inform you that your manuscript has been judged scientifically suitable for publication and will be formally accepted for publication once it meets all outstanding technical requirements.

Kind regards,

Billy Morara Tsima, MD MSc

Academic Editor

PLOS ONE
---

## [Editor Report · Acceptance letter]

PONE-D-23-25640R3

PLOS ONE

Dear Dr. Iizuka,

I'm pleased to inform you that your manuscript has been deemed suitable for publication in PLOS ONE. Congratulations! Your manuscript is now being handed over to our production team.

Kind regards,

on behalf of

Dr. Billy Morara Tsima

Academic Editor

PLOS ONE